# A CRISPR-Cas12a-derived biosensing platform for the highly sensitive detection of diverse small molecules

Mindong Liang[1,2,13], Zilong Li[3,13], Weishan Wang [3,13], Jiakun Liu[1,4,13], Leshi Liu[1], Guoliang Zhu[1], Loganathan Karthik[1], Man Wang[5], Ke-Feng Wang[1], Zhong Wang[6], Jing Yu[7], Yuting Shuai[1], Jiaming Yu[1], Lu Zhang[1], Zhiheng Yang[1], Chuan Li[1], Qian Zhang[1], Tong Shi[1], Liming Zhou[1], Feng Xie [3], Huanqin Dai[3], Xueting Liu[1], Jingyu Zhang[1], Guang Liu [1], Ying Zhuo[1], Buchang Zhang[2], Chenli Liu[4], Shanshan Li[8], Xuekui Xia[9], Yaojun Tong [10], Yanwen Liu[11], Gil Alterovitz[12], Gao-Yi Tan [1] & Li-Xin Zhang[1]

Besides genome editing, CRISPR-Cas12a has recently been used for DNA detection applications with attomolar sensitivity but, to our knowledge, it has not been used for the detection of small molecules. Bacterial allosteric transcription factors (aTFs) have evolved to sense and respond sensitively to a variety of small molecules to benefit bacterial survival. By combining the single-stranded DNA cleavage ability of CRISPR-Cas12a and the competitive binding activities of aTFs for small molecules and double-stranded DNA, here we develop a simple, supersensitive, fast and high-throughput platform for the detection of small molecules, designated CaT-SMelor (CRISPR-Cas12a- and aTF-mediated small molecule detector). CaT-SMelor is successfully evaluated by detecting nanomolar levels of various small molecules, including uric acid and p-hydroxybenzoic acid among their structurally similar analogues. We also demonstrate that our CaT-SMelor directly measured the uric acid concentration in clinical human blood samples, indicating a great potential of CaT-SMelor in the detection of small molecules.

[1] State Key Laboratory of Bioreactor Engineering, and School of Biotechnology, East China University of Science and Technology (ECUST), 200237 Shanghai, China. [2] Institute of Physical Science and Information Technology, Anhui University, 230601 Hefei, China. [3] State Key Laboratory of Microbial Resources and CAS Key Laboratory of Pathogenic Microbiology and Immunology, Institute of Microbiology, Chinese Academy of Sciences, 100101 Beijing, China. [4] Shenzhen Institute of Synthetic Biology, Shenzhen Institutes of Advanced Technology, Chinese Academy of Sciences, 518055 Shenzhen, China. [5] School of Agriculture and Biology, Shanghai Jiao Tong University, 200240 Shanghai, China. [6] Department of Urology, Sixth People's Hospital South Campus Affiliated to Shanghai Jiao Tong University, 201499 Shanghai, China. [7] Department of Clinical Laboratory, Sixth People's Hospital South Campus Affiliated to Shanghai Jiao Tong University, Shanghai 201499, China. [8] State Key Laboratory for Biology of Plant Diseases and Insect Pests, Institute of Plant Protection, Chinese Academy of Agricultural Sciences, 100193 Beijing, China. [9] Key Biosensor Laboratory of Shandong Province, Biology Institute, Qilu University of Technology (Shandong Academy of Sciences), 250013 Jinan, China. [10] The Novo Nordisk Foundation Center for Biosustainability, Technical University of Denmark, 2800 Kgs Lyngby, Denmark. [11] inBiome Sciences, 3403 SW 86th ST, Gainesville, FL 32608, USA. [12] Harvard Medical School Countway Library, 10 Shattuck Street, Boston, MA 02115, USA. [13] These authors contributed equally: Mindong Liang, Zilong Li, Weishan Wang, Jiakun Liu. Correspondence and requests for materials should be addressed to G.-Y.T. (email: tangy@ecust.edu.cn) or to L.-X.Z. (email: lxzhang@ecust.edu.cn)

 **1**

During evolution, a plethora of bacterial allosteric transcription factors (aTFs) have been optimized to sense and respond to a variety of small molecules[1]. aTFs typically comprise a DNA-binding domain and an effector-binding domain[2,3]. The presence and binding of the target molecule usually induces a conformational change in the aTF DNA-binding domain, which then enhances or attenuates the double-stranded DNA (dsDNA)-binding capacity of the aTF by the direct binding of the aTF effector-binding domain[1]. Recently, we developed a bacterial-aTF-based in vitro biosensing platform, aTF-NAST (aTF-based nicked DNA-template-assisted signal transduction) by exploiting the competition between T4 DNA ligase and aTFs in binding to nicked DNA[4,5]. However, this is a relatively time-consuming, inconvenient and costly method of detecting small molecules.

CRISPR-Cas12 (Cpf1) proteins are RNA-guided enzymes that bind and cleave DNA, as components of bacterial immune systems[6,7]. Unlike CRISPR/Cas9 proteins, the Cas12a endonucleases only require CRISPR RNA (crRNA), but not *trans*-activating crRNA (tracrRNA), as their guide. They recognize a T-rich protospacer-adjacent motif (PAM) instead of a G-rich PAM and generate dsDNA breaks with staggered 5′ ends[6]. To date, CRISPR/Cas12a-based tools have been widely used in many areas, including genome manipulation[6,8], DNA assembly[9,10], DNA steganography[11], and so on. The *trans* cleavage activity of CRISPR-Cas12a on single-stranded DNA (ssDNA) has also recently been reported[12,13]. This unique property of Cas12a has since been applied to the detection of nucleic acids and offers a strategy for improving the specificity, sensitivity, and speed of nucleic acid-based diagnostic applications[13,14]. However, Cas12a itself seems incapable of detecting small molecules (or molecules with low molecular weights) in fields such as disease diagnosis, food quality control, environmental pollutant detection and metabolic engineering[15,16].

Here, we develop a small molecule detection platform that combines CRISPR-Cas12a and aTF, designated CaT-SMelor (<u>C</u>RISPR-Cas12<u>a</u> and a<u>T</u>F mediated <u>s</u>mall <u>m</u>olecule detector). As a proof of concept, different kinds of small molecules are selected, such as those involved in the diagnosis of metabolic diseases, antibiotic residues, and food preservative, to demonstrate the successful setting of our CaT-SMelor. Clinical human serum samples are also tested for uric acid (which causes gout)[17]. Our results indicate that the methodology we have developed in this study has great potential in the detection and measurement of diverse small molecules for many different purposes.

## Results

**Design of CaT-SMelor**. A flow chart of the action of CaT-SMelor is shown in Fig. 1. Briefly, an aTF fused to a cellulose-binding domain (CBD–aTF) was immobilized on microcrystalline cellulose. The functional dsDNA, which contained both the PAM sequence and the binding motif of the corresponding aTF, specifically binds the DNA-binding domain of the aTF. In the presence of the target small molecule, the conformation of the aTF changed, leading to the dissociation of dsDNA from the aTF-binding domain. The free dsDNA then bound to the Cas12a–crRNA complex, activating the nonspecific ssDNA *trans* cleavage activity of Cas12a (activated Cas12a). Activated Cas12a starts cleaving a fluorophore quencher (FQ)-labeled ssDNA probe[18]. Target small molecules can be qualitatively or quantitatively analyzed by measuring the change in the fluorescent signal.

**Evaluation of selected CBD–aTFs allosteric activities**. Because both aTFs and the Cas12a–crRNA complex can bind to the same

target dsDNA, the presence of both aTF and Cas12a in the same buffer system allows them to compete for and interfere with the same target dsDNA. Therefore, we engineered the aTFs so that they could be immobilized to avoid such potential interference. We fused the aTFs to a full-length CBD with a flexible 20-amino-acid linker (SGGGG)$_4$[19] (Fig. 2a). All the engineered CBD–aTF recombinants used in this study were successfully expressed and purified with a histidine (His) tag (Supplementary Fig. 1).

We then determined whether the engineered CBD–aTFs were still fully functional. An electrophoretic mobility shift assay (EMSA) was used to evaluate the dsDNA-binding activity and allosteric activity of the CBD–aTFs. We measured the dsDNA (containing the HucR binding motif)-binding affinity of 38-kDa CBD-HucR. As the protein concentration increased from 25 to 200 nM, an increasing shift in the dsDNA (CBD–HucR–dsDNA complex) was observed (Fig. 2b, Supplementary Fig. 2A).

In contrast, when we added gradually increasing concentrations (0.05–500 μM) of uric acid, the effector of HucR[20], to the system, increasing amounts of dsDNA dissociated from the CBD–HucR–dsDNA complex (Fig. 2c, Supplementary Fig. 2B). We also evaluated other CBD–aTFs, such as CBD-TetR and CBD-HosA, with their paired effectors (tetracycline[21] and food preservative *p*-hydroxybenzoic acid [*p*-HBA][22,23], respectively). Both CBD-TetR and CBD-HosA show perfect dsDNA-binding and effector-induced allosteric activities (Fig. 2d–g, Supplementary Fig. 3–4). These results indicated that the full functions of the engineered aTFs were preserved. Furthermore, when we compared the dsDNA-binding and allosteric activities of the original HosA (Supplementary Fig. 5), there were no significant differences between HosA and CBD-HosA on gel-shift images or in the percentage of dsDNA shifted (Fig. 2d, e, Supplementary Fig. 3). These results indicated that the fusion of CBD had no significant effect on the functions of the selected aTFs.

**CaT-SMelor discriminates target small molecules from analogs**. A brief flow chart of the process by which CaT-SMelor senses and detects small molecules is shown in Fig. 3a. In this experimental set-up, the dsDNA containing the specific aTF-binding motif and PAM sequence were bound to the immobilized CBD–aTF. The superfluous free dsDNA was removed by centrifugation and washing with fresh buffer. In the second step, the presence of the effector induced the dissociation of the dsDNA from the CBD–aTF–dsDNA complex, and the resulting free dsDNA in the supernatant was collected. This dsDNA was added to the CRISPR/Cas12a-containing system to activate the *trans* cleavage activity of Cas12a. Therefore, the FQ-labeled ssDNA in the system was cleaved and the fluorescent signal was detected with a fluorescence reader.

A variety of purine compounds (guanine, adenine, and hypoxanthine; Supplementary Fig. 6) and uric acid were selected to test the specificity of CaT-SMelor. Uric acid generated the strongest fluorescent signal. The fluorescent signals generated by the other three molecules (guanine, adenine, and hypoxanthine) were <5% of the control signal (Fig. 3c). These results were consistent with the EMSA results shown in Fig. 3b and Supplementary Fig. 7. *p*-HBA and its structurally similar analogs (Supplementary Fig. 8) were also used to test CaT-SMelor. Even in the presence of structurally similar analogues at final concentrations of 1.8 mM, negligible dissociation of dsDNA and negligible fluorescent signal were detected. However, *p*-HBA induced the almost complete dissociation of dsDNA from the CBD–HosA–dsDNA complex, generating the maximum fluorescent signal (Fig. 3e). These results were also consistent with the EMSA results (Fig. 3d, Supplementary Fig. 9), and indicated that CaT-SMelor accurately responded to dissociated dsDNA (signal

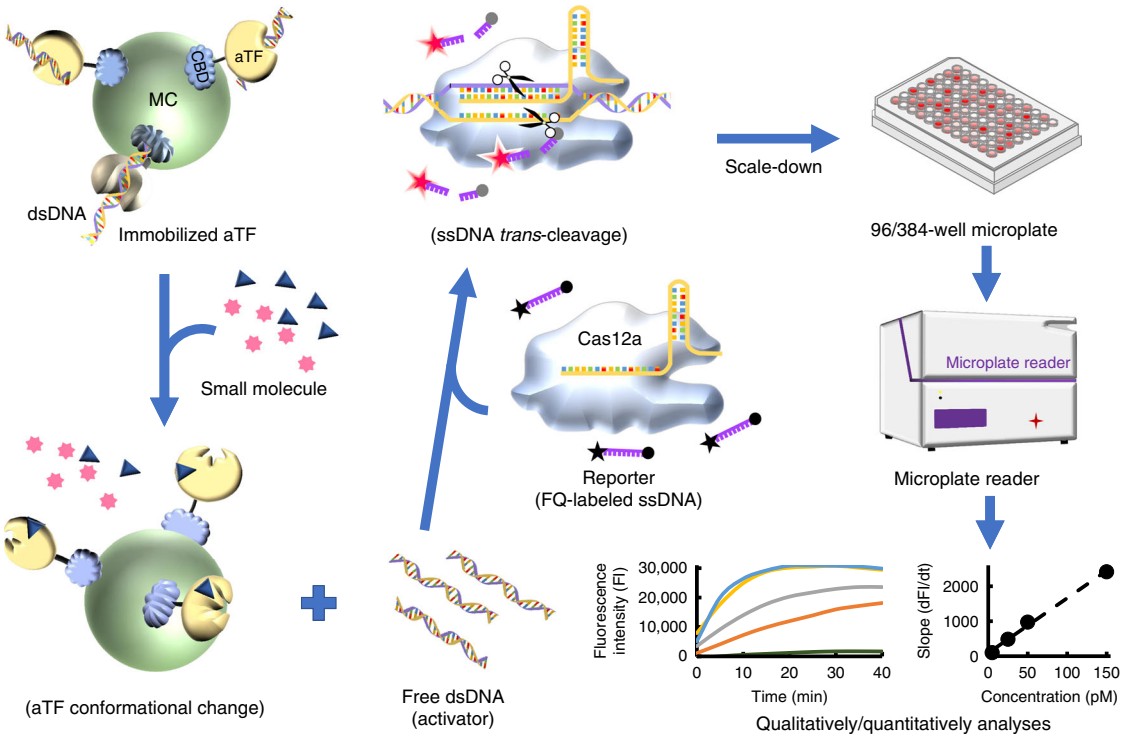

**Fig. 1** Schematic diagram of the CaT-SMelor. MC microcrystalline cellulose, aTF allosteric transcription factor, CBD cellulose-binding domain, dsDNA double-stranded DNA, FQ-labeled ssDNA fluorophore-quencher-labeled single-stranded DNA

input) and discriminated small molecule targets with a fluorescent signal (signal output).

**CaT-SMelor quantitatively detects small molecule targets**. The principle underlying this method is that the unreadable signals of the small molecule targets are transformed into easily readable dsDNA signals by aTFs. By coupled with CRISPR-Cas12a and an FQ-labeled reporter system, the dsDNA signal is detected with high sensitivity in a relatively short time. In this study, both $dsDNA_{(TetR)}$ and $dsDNA_{(HucR)}$ were used to test the sensitivity and linear range of CaT-SMelor (Supplementary Table 5). As little as several pM concentrations of $dsDNA_{(TetR)}$ or even an fM concentration of $dsDNA_{(HucR)}$ activated the ssDNA-trans-cleavage activity of Cas12a. As the activator dsDNA concentrations and the cleavage time were increased, the fluorescent signal gradually increased (Fig. $4a_{(L)}$, Supplementary Fig. 10A). The fluorescent signal increased linearly within 15–20 min of cleavage, with $R^2 > 0.99$ (Supplementary Fig. 10A, 11A). When we examined the relationship between the slope of the fluorescence intensity (e.g., cleavage rate) and the dsDNA concentration (Supplementary Fig. 10B, 11B), the cleavage rate shows an excellent linear pattern ($R^2 > 0.99$) as the dsDNA concentration increased in the range 5–150 pM ($dsDNA_{(HucR)}$) or 1–25 pM ($dsDNA_{(TetR)}$) (Fig. $4a_{(R)}$, Supplementary Fig. 11C).

Once again, we used uric acid and p-HBA to demonstrate the effectiveness of CaT-SMelor in a quantitative analysis. After incubation with different concentrations of uric acid, the free dsDNA signal in the supernatant was detected with the Cas12a-based FQ-labeled reporter system. The results indicated that the limit of detection for uric acid was 10 nM (final concentration), and the fluorescent signal gradually increased as the concentration of uric acid increased. The slope of the fluorescence intensity (e.g., cleavage rate) was linear ($R^2 > 0.99$) as the uric acid concentration increased in the range 25–500 nM (Fig. 4b, Supplementary Fig. 12). In Fig. 4c and Supplementary Fig. 13,

p-HBA concentrations as low as 1.8 nM were also sensitively detected with CaT-SMelor, with a quantitative analysis of p-HBA possible in a concentration range 9–180 nM. These results suggested that CaT-SMelor quantitatively detected small molecule targets with high sensitivity.

**CaT-SMelor can be used for clinic analysis of blood samples**. The uric acid in human clinical serum samples was analyzed very accurately with high-performance liquid chromatography (HPLC) (Supplementary Fig. 14). At the same time, the concentrations of uric acid in the same serum samples were analyzed with CaT-SMelor (Supplementary Fig. 15). The concentrations of uric acid in each sample determined with the two different methods were directly proportional ($y = 0.996x$, $R^2 = 0.9895$) (Fig. 5a, Supplementary Fig. 16). We also analyzed the uric acid in the serum samples with a clinical automatic biochemical analyzer system. The results for each sample obtained with CaT-SMelor and the biochemical analyzer were almost identical ($y = 1.0374x$, $R^2 = 0.9838$) (Fig. 5b, Supplementary Fig. 16). Unlike routine analytical methods, CaT-SMelor does not rely on a delicate device, so under certain circumstances, it should be more applicable than conventional methods.

The normal range of uric acid for human was 2.8–9.2 mg dl$^{-1}$ (166.4–546.7 μM)[24]. Because CaT-SMelor is highly sensitive, very tiny amounts of blood can be analyzed for uric acid. As shown in Fig. 5c, fresh blood samples (~1 μL) were used directly for the analysis of uric acid with CaT-SMelor (Supplementary Table 1). CaT-SMelor required only 50% of the time required by current clinical laboratory techniques for the analysis of serum (Supplementary Fig. 17). Obviously, CaT-SMelor is clearly a cheaper, more efficient, and more user-friendly method of analyzing blood uric acid (Table 1).

**The advantages of CaT-SMelor over other selected methods**. The comparison between our CaT-SMelor method and other

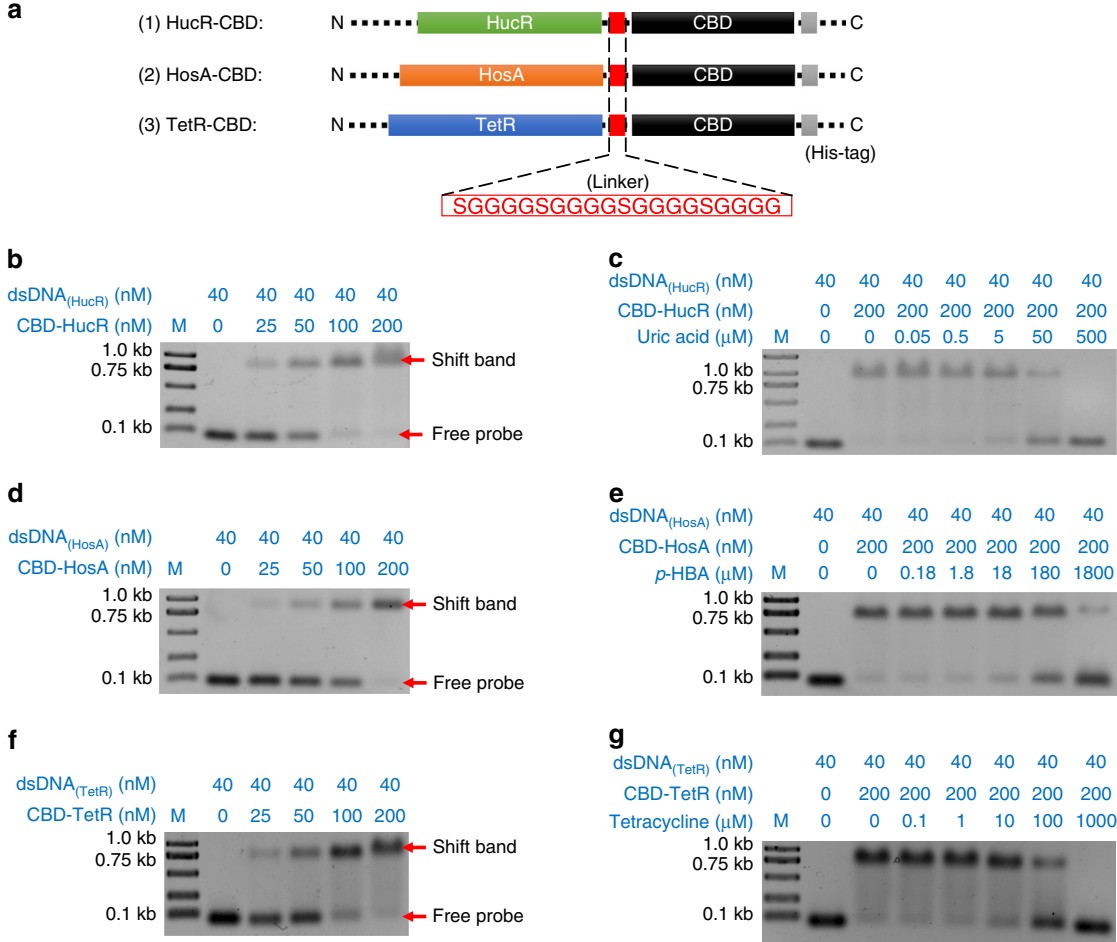

**Fig. 2** Evaluation of the activities of CBD–aTFs with EMSA. **a** Schematic diagram of CBD–aTF. **b** EMSA with CBD-HucR and dsDNA containing the HucR-binding motif; M: 1 kb DNA ladder marker. **c** EMSA evaluation of CBD-HucR allosteric activities in the presence of uric acid. **d** EMSA of CBD-HosA and dsDNA containing the HosA-binding motif. **e** EMSA evaluation of CBD-HosA allosteric activity in the presence of p-hydroxybenzoic acid (p-HBA). **f** EMSA of CBD–TetR and dsDNA containing the TetR-binding motif. **g** EMSA evaluation of CBD–TetR allosteric activity in the presence of tetracyclines. Relative quantitative analysis of EMSA data is shown in Supplementary Fig. 2–4

several previously reported small molecules detection methods was presented in Table 1. From the comparison, we concluded that the CaT-SMelor method performed very nicely in small molecules detection. It had many advantages over other methods, such as convenient operation, low limit of detection, and available for 96-well and 384-well workstations. As the fluorophore signal is relatively stable, the half-time of fluorescence in CaT-SMelor is more than 200 min (Supplementary Fig. 18), which indicated that users would have more time for precise results recording. In addition, the whole determination process (including incubation) could be finished in at most 25 min. And if the concentration of target small molecules, e.g. uric acid, in the sample is higher than 250 nM, the *trans*-cleavage of CRISPR-Cas12a and subsequently fluorescence detection process could be shortened to 5–10 min (Fig. 3a). Most importantly, by a carefully cost accounting of per reaction, the CaT-SMelor method was totally affordable for high throughput assay (Supplementary Table 2).

## Discussion
The CRISPR/Cas13-based and CRISPR/Cas12a-based detection platforms have shown remarkable advantages for nucleic-acid detection, including its sensitivity, accuracy, speed, etc[25–27]. In the present study, our initial idea was that the CRISPR/Cas-mediated nucleic-acid detection platform could be regarded as an excellent general signal output and display module. Coupling it to

an appropriate signal input module might significantly improve the performance of the detection system. In this study, an aTF-based in vitro small-molecule-biosensing platform was coupled to the CRISPR/Cas12-mediated nucleic-acid-detection system, and the small molecule could be rapidly and sensitively detected (Fig. 1). This system constitutes CaT-SMelor. CaT-SMelor was subsequently shown to be efficient and highly accurate in the rapid analysis of small molecules, as expected.

Bacterial aTFs are widespread transcription regulatory proteins, most of which can sense various small molecules, including metabolites, chemicals, heavy metals, etc[28]. To date, many naturally occurring aTFs, such as HucR, HosA, BgaR, and TetR, have been well documented and used in the detection of small molecules[2–5,29]. In this study, our preliminary data show that dsDNA that contains both the aTF-binding motif and the PAM sequence more readily bound Cas12a other than aTFs, probably because the binding affinity between the Cas protein and the PAM sequence was higher than that between the aTF and the aTF-binding motif[30], especially in the presence of crRNA. Therefore, the whole process can be divided into three steps: (1) immobilization of aTF (or an aTF-dsDNA complex); (2) incubation of the targeted small molecules with the aTF-dsDNA complex and then the removal of the immobilized aTF by centrifugation; and (3) detection of free dsDNA by the CRISPR/Cas12a-based FQ-labeled reporter system (Figs. 1 and 3a). Based

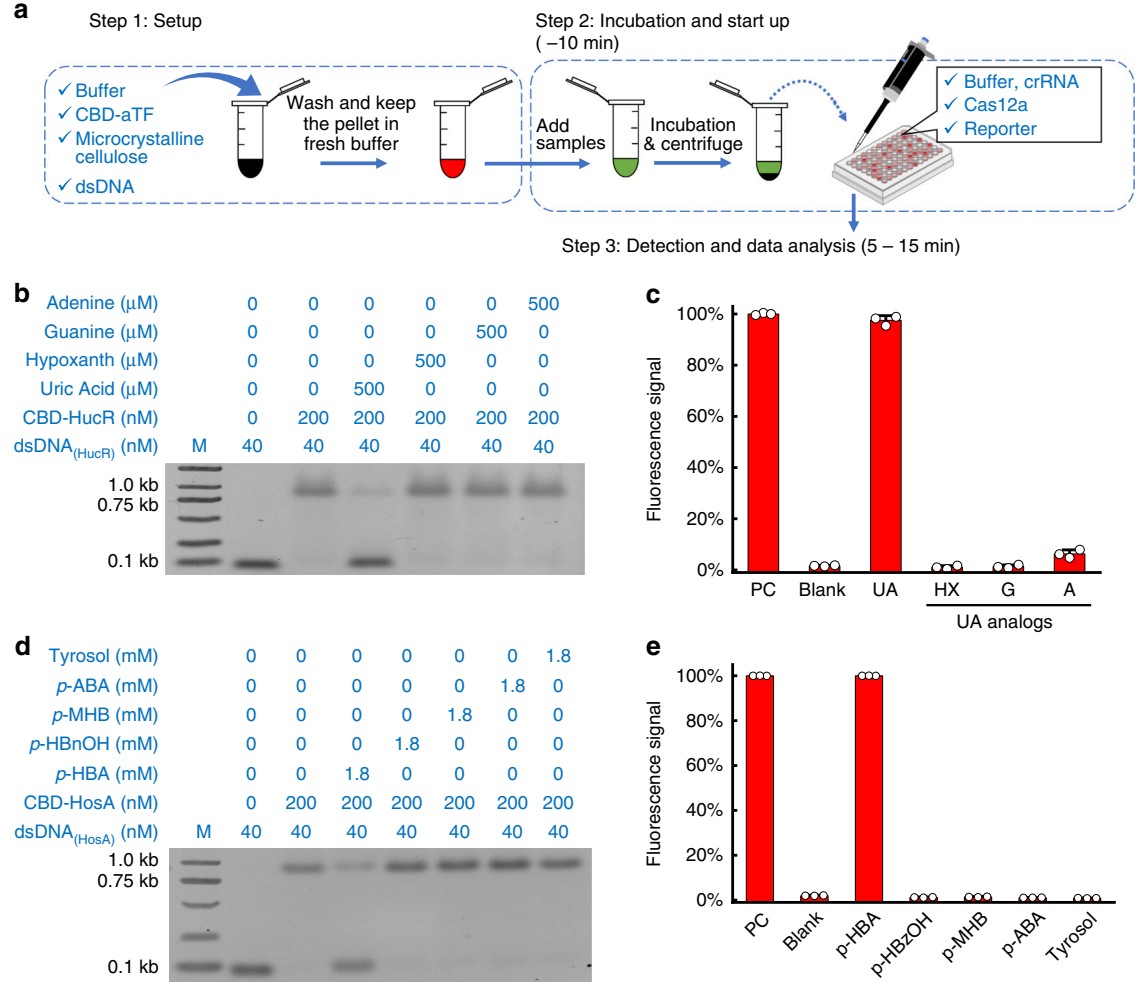

**Fig. 3** Detection of target small molecules by CaT-SMelor. **a** Flow chart of the sensing and detection of small molecules by CaT-SMelor. **b** Dissociation of dsDNA from the CBD–HucR–dsDNA $_{(HucR)}$ complex in the presence of uric acid and its structural analogues. **c** Fluorescence intensity of CaT-SMelor in the presence of 500 μM uric acid, hypoxanthine (HX), guanine (G), or adenine (A). Superfluous free dsDNA $_{(HucR)}$ was used as the positive control (PC); M: 1 kb DNA ladder marker. **d** Dissociation of dsDNA $_{(HosA)}$ from the CBD–HosA–dsDNA $_{(HosA)}$ complex in the presence of p-hydroxybenzoic acid and its structural analogues. **e** Fluorescence intensity of CaT-SMelor in the presence of 1.8 mM p-hydroxybenzoic acid (p-HBA), p-hydroxybenzyl alcohol (p-HBnOH), methyl-p-hydroxybenzoate (p-MHB), p-aminobenzoic acid (p-ABA), or tyrosol. Superfluous free dsDNA $_{(HosA)}$ was used as the positive control (PC). Error bars are means and SDs from three independent repeats. The corresponding dot plots are also presented in the bar charts

on this work flow, some well-characterized aTF proteins, such as HosA and HucR, were then used to demonstrate the effectiveness of detection platform.

Protein immobilization via the CBD has been demonstrated previously[31,32]. The correct protein folding and structure of an aTF are prerequisites for its sensing ability. Therefore, in this study, we investigated the effects of CBD fusion on the activity of aTFs. The results indicated that the fusion of the 17-kD CBD to HucR, HosA, or TetR had no detrimental effect on their dsDNA-binding and allosteric activities (Fig. 2, Supplementary Fig. 2–5). CBD probably acts as a mediator in the immobilization of the protein, adsorbing it to the microcrystalline cellulose. It is an economic and highly efficient method of aTF immobilization.

The performance of CaT-SMelor in detecting types of small molecules was then tested. It should be noted that the specificity of CaT-SMelor relies on the aTF used. Using well-characterized CBD-HucR and CBD-HosA[3,5], uric acid and p-HBA were qualitatively detected with CaT-SMelor, respectively (Fig. 3b–e). We also found that CBD-TetR only detected the tetracyclic structure of tetracyclines, and various kinds of tetracyclines, such as chlortetracycline and deoxytetracycline, were detected (data not

shown). The currently recognized aTFs are naturally occurring proteins. Because the binding affinities between aTFs and their effectors differ, naturally occurring aTFs may not always be satisfactory for use with CaT-SMelor. However, if necessary, protein libraries could be screened for more suitable aTFs or candidate aTFs could be engineered with directed evolution technologies, such as phage-assisted continuous evolution[33].

After its activation by dsDNA, Cas12a-crRNA catalyzes *trans* ssDNA cleavage with very high catalytic efficiency, which approaches the rate of diffusion[13,34]. We also found that the *trans* ssDNA cleavage activity of CRISPR-Cas12a is activated by dsDNA concentrations as low as ~fM (Fig. 4a, Supplementary Fig. 10–11). Consistent with the Michaelis–Menten plot for the CRISPR/Cas12a-catalyzed *trans* cleavage of ssDNA using a dsDNA activator[13], with the increase in dsDNA in the appropriate concentration range, the *trans* ssDNA cleavage activity of CRISPR-Cas12a also increased in a linear manner ($R^2 > 0.99$). Nanomolar concentrations of both uric acid and p-HBA were also quantitatively analyzed with CaT-SMelor (Fig. 4b, c). The results were consistent with the EMSA data shown in Fig. 2 and the dissociation constants ($K_d$) of these aTF–dsDNA complexes[2,20].

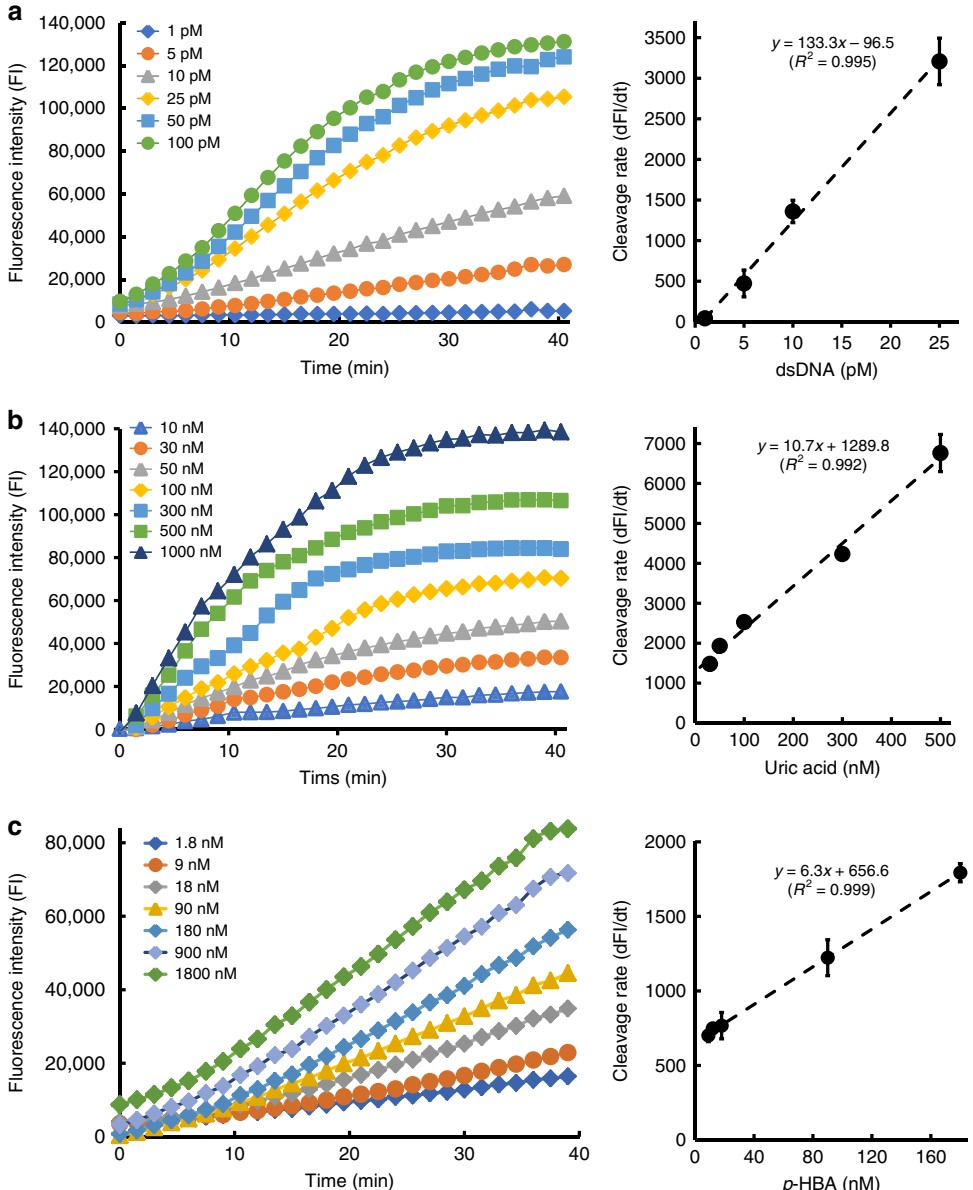

**Fig. 4** Quantitative analysis of small molecules by CaT-SMelor. **a** Effects of different concentrations of dsDNA(activator) on the ssDNA *trans* cleavage activity of CRISPR-Cas12a (left). The linear relationship between the activator dsDNA concentration and the cleavage rate (slope of the fluorescence intensity, dFI/dt) of CRISPR-Cas12a (right). The concentration range in which the calibration curve was linear was 1–25 pM ($R^2 = 0.995$). **b** Readouts of fluorescence intensity in the presence of different concentrations of uric acid (left). The linear relationship between the concentration of uric acid and the cleavage rate (right). The concentration range in which the calibration curve was linear was 25–500 nM ($R^2 = 0.992$). **c** Readout of fluorescence intensity in the presence of different concentrations of *p*-HBA (left). The linear relationship between the concentration of *p*-HBA and the cleavage rate (right). The concentration range in which the calibration curve was linear was 9–180 nM ($R^2 = 0.999$). The background signal (control) has already been subtracted from the values displayed in the graph. Error bars are means and SDs from at least two independent repeats. Source data are available in the Source Data file

Moreover, it also should be noted that the linear detection range of CaT-SMelor for targeted small molecules can be tuned by changing the sequence of the dsDNA[2].

To evaluate the validity and clinical applications of CaT-SMelor, we used it to analyze uric acid in human blood samples. Human serum not only contains large amounts of proteins[35], but also has a very complex chemical composition, and is widely used for disease diagnosis. In this study, serum uric acid was detected as a biomarker of chronic gout[36]. The uric acid concentrations estimated with a Beckman coulter biochemical analyzer system and HPLC were highly consistent ($R^2 > 0.99$) with those determined with CaT-SMelor (Fig. 5a, b). The high sensitivity of CaT-SMelor means that as little as 1 μL of blood is sufficient for the analysis of uric acid, and can be used directly, with no sample preparation (Fig. 5c). CaT-SMelor has many unique advantages over previous aTF-based and protein-immobilization-assisted methods of small-molecule detection[4,5], (Table 1) and has great potential utility in the future development of a miniature portable device for the detection of small molecules or disease diagnoses.

Based on the results described above, we concluded that aTFs and the CRISPR/Cas12-mediated nucleic-acid-detection system could be efficiently coupled for the qualitative and quantitative analysis of small molecules by protein immobilization. CaT-SMelor, described here, also offers a strategy for developing a

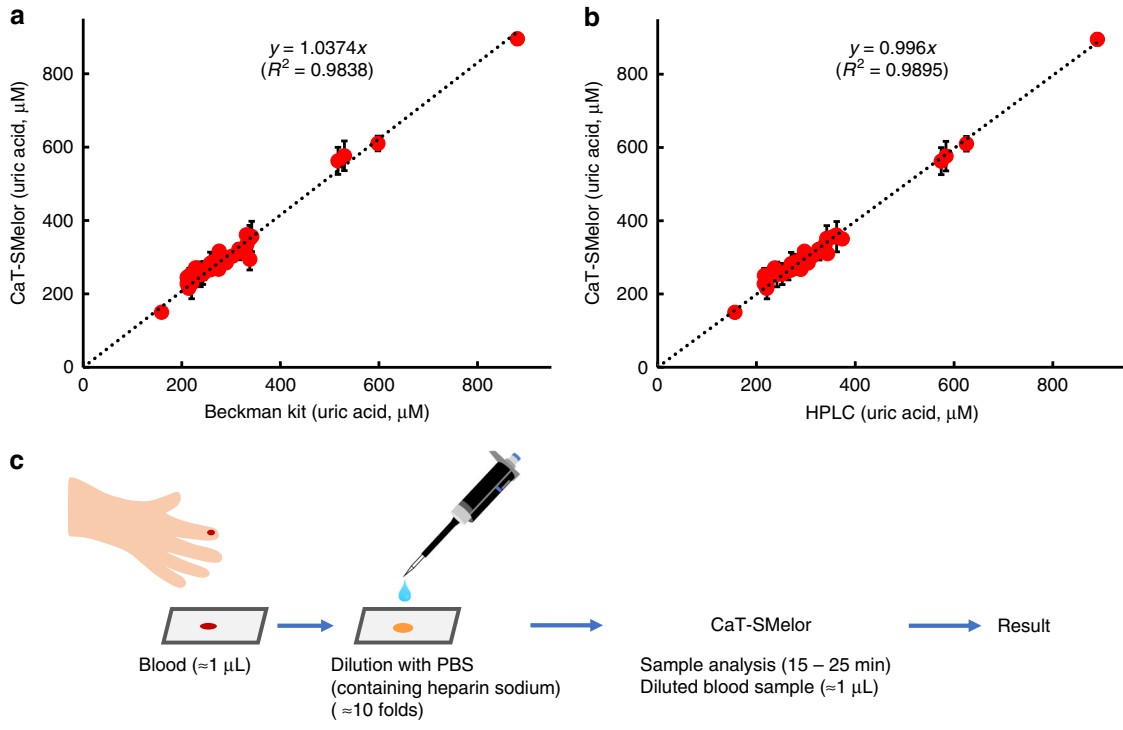

**Fig. 5** Analysis of uric acid concentration in human blood samples. Uric acid was detected in 32 random human serum samples. **a** Analysis of uric acid with HPLC and CaT-SMelor. **b** Analysis of uric acid with a clinical automatic biochemical analyzer system and CaT-SMelor. **c** Flow chart of CaT-SMelor used for blood sample analysis. Error bars are means and SDs from at least two independent repeats. Source data are available in the Source Data file

**Table 1 Comparison of different methods of small-molecule detection**

| Method | LOD | AOM (Y/N) | TC[a] | Cost[b] | Ref. |
|---|---|---|---|---|---|
| qPCR | ~ nM | Y | >120 | >2 | 4 |
| RCA | ~ nM | Y | >120 | NA | 4 |
| RPA | ~ pM | N | >120 | >100 | 4 |
| PGM | ~ nM | N | >15 | NA | 39 |
| ISDA | ~ nM | Y | >120 | NA | 3 |
| Clinical method[c] | ~ μM | Y | >30 | >2 | This study |
| HPLC | ~ μM | N | >25 | NA | This study |
| CaT-SMelor | ~ nM | Y | 15–25 | <0.3 | This study |

*LOD* limit of detection, *AOM* availability of microplate, *TC* time consuming, *qPCR* quantitative PCR, *RCA* rolling circle amplification, *RPA* recombinase polymerase amplification, *PGM* personal glucose meter, *ISDA* isothermal strand displacement amplification
[a]min/reaction
[b]$/reaction
[c]Uric acid reagent (Kit number 442785, Beckman Coulter, Inc.)

Triple-3S (**s**imple, **s**ensitive, and **s**peed) inexpensive high-throughput method for the analysis of small molecules.

## Methods

**Plasmid construction and protein purification**. The plasmids used in this study are listed in Supplementary Table 3. *Lachnospiraceae* bacterium ND2006 (LbCas12a) has been used previously for dsDNA and ssDNA cleavage[6]. The CBD-domain-encoding gene was amplified from pET35b and then cloned into the *Xho*I site of the pET28a with the EZmax one step seamless cloning kit (Tolo Biotechnology, Shanghai). All the primers used for plasmid construction are listed in Supplementary Table 4. For proteins purification, the cell pellets were resuspended in lysis buffer [50 mM Tris-HCl (pH 7.4), 200 mM NaCl, 2 mM dithiothreitol (DTT), 5% glycerol] supplemented with protease inhibitors (Roche, EDTA-free), and then lysed with ultrasonication. The lysate was loaded onto a HisTrap FF

column (GE Healthcare) and washed with a gradient of imidazole concentrations. The peak fractions were collected and desalted with dialysis. The solution was then loaded onto a HiTrap Q HP column (GE Healthcare), and the peak fractions were collected and concentrated. The concentrated solution was loaded onto a HiLoad 16/600 Superdex 200 pg column for fast protein liquid chromatography (FPLC; AKTA Explorer 100, GE Healthcare). The gel filtration fractions were analyzed with SDS-PAGE and the concentrations were determined with the Bradford method.

**Oligonucleotide and crRNA preparation**. The oligonucleotides used as templates for crRNA transcription and the primers used for DNA amplification were synthesized by GenScript (Supplementary Table 5). crRNA was prepared with in vitro transcription. To prepare the templates for crRNA synthesis, two paired primers containing a T7 priming site were synthesized and annealed in *Taq* DNA Polymerase PCR Buffer (Thermo Fisher Scientific). The crRNAs were then transcribed with the HiScribe™ T7 Quick High Yield RNA Synthesis Kit (NEB) and purified with RNA Clean & Concentrator™-5 (Zymo Research). The resulting crRNA was quantified with a NanoDrop 2000 spectrophotometer (Thermo Fisher Scientific). Throughout the experiments, RNase free materials and conditions were applied.

**Electrophoretic mobility shift assay (EMSA)**. The probes for the EMSAs were amplified with PCR and purified with the Axygen PCR Cleanup Kit (Supplementary Table 5). Each probe was incubated at room temperature for 30 min with different concentrations (0, 25, 50, 100, and 200 nM) of purified CBD–aTF protein in 20 μL of binding buffer [10 mM Tris–HCl (pH 7.5), 100 mM KCl, 1 mM EDTA, 0.1 mM DTT, 5% v/v glycerol, 0.01 mg mL$^{-1}$ bovine serum albumin][37,38]. For the dissociation assays, different concentrations of small molecules and different kinds of structural analogues were added to the binding reaction mixtures. After incubation, the mixtures were separated in 1.5% agarose gel with Tris-borate-EDTA buffer at 4 °C. The band shifts were detected and analyzed with the ChemiDoc XRS Gel Imaging System (Bio-Rad).

**Protein immobilization and small-molecule sensing**. Microcrystalline cellulose (CAS No. 9004-34-6) was supplied by Shanghai Sangon Biotech (China). After the microcrystalline cellulose was washed twice with Tris-HCl buffer, 2 mg was mixed with the purified CBD–aTF protein (200 nM) in NEB CutSmart® Buffer and then incubated at room temperature for ~10 min. The superfluous free protein (not immobilized) was removed by washing with NEB CutSmart® Buffer. The immobilized protein was quantified with a Bradford assay kit (Tiangen, China). The

immobilized aTF proteins were incubated with dsDNA (100 nM) containing the binding motif of the corresponding aTF for 8–10 min. The superfluous free (unbound) dsDNA was removed by washing. The mixture of immobilized CBD–aTF–dsDNA was then distributed in a 384-well plate (or Eppendorf tubes). For small molecule sensing, 1–10 µL of different concentrations of small molecules (e.g., uric acid, p-HBA) were added to the corresponding wells. After a short incubation, the supernatants were used for the subsequent experiment.

**ssDNA trans cleavage and florescence detection**. In the CRIPSR-Cas12 cutting system, NEB CutSmart® Buffer was adapted as the reaction buffer (Supplementary Table 2). For the *trans* cleavage assay, equal molar ratios of LbCas12a and crRNA were premixed with the FQ-labeled reporter in NEB CutSmart® Buffer and then distributed in 384-well plates for the subsequent experiment. Samples containing the activator dsDNA were added to the reporter system, and the florescent signal was detected with a BMG CLARIOstar microplate reader (BMG Labtech, UK) at an excitation wavelength of 480 nm and emission wavelength of 520 nm.

**Analysis of clinical human serum samples**. The use of human serum samples for this study was approved by Ethics Committees of Sixth People's Hospital South Campus Affiliated to Shanghai Jiao Tong University, and complied with all relevant ethical regulations. All participants provided informed consent for blood donation. Uric acid reagent (Beckman Coulter, Inc.) and an automatic biochemical analyzer (Unicel DxC 800 Synchron Clinical System; Beckman Coulter) were used to analyze clinical serum samples for uric acid. Equal volumes of serum and chloroform were mixed and vortexed vigorously to remove most proteins, and the supernatant was collected by centrifugation at $10,000 \times g$ for 10 min. The sample was then used for HPLC analysis. Uric acid was detected with an Agilent SB-Aq column (4.6 mm × 150 mm, 5 µm, 1 mL min⁻¹) with $H_2O$ as the eluent and detection at 284 nm with a UV detector (Agilent 1260 Infinity II LC System). For the CaT-SMelor analysis, to avoid any pipetting error, the blood samples were diluted 10–20-fold with phosphate-buffered saline (or water), and 1–2 µL of the diluted serum was used directly for the uric acid analysis. For calibration curve construction, after incubation with 25–500 nM of uric acid standard, the free dsDNA in the supernatant, which dissociated from HucR (immobilized)-dsDNA (TAGGTAGACATCTAAGTA, 5′–3′) complex, were detected with the Cas12a-based FQ-labeled reporter system. The time course of fluorescence intensity changes was measured (BMG CLARIOstar microplate reader, BMG Labtech, UK), and the slope of fluorescence linear region between 5 and 20 min (normally) represents a ssDNA trans-cleavage rate of CRISPR/Cas12a. The linear relationship between trans-cleavage rate (or slope) and uric acid standard concentration was obtained.

**Date analysis**. Date analysis and process were carried out using Microsoft Excel 2016. The data in figures were expressed as mean ± standard deviation (SD).

**Reporting summary**. Further information on research design is available in the Nature Research Reporting Summary linked to this article.

## Data availability

The source data underlying Figs. 3C, 3E, 4A, 4B, 4C, 5A, and 5B are provided as a Sources Data file. All other data are available from the corresponding author (G.-Y.T or L.-X. Z) authors upon reasonable request.

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

## Acknowledgements

The authors would like to thank Xian'en Zhang, Jianguo Shi, Huarong Tan, Changheng Liu, Pinghua Liu, Yihua Chen, Minyong Li, Linquan Bai, Jian-Jiang Zhong, Tiangang Liu and Shi Chen for helpful discussions. This work was supported by the National Natural Science Foundation of China [31430002, 31720103901, 31870040, 21877038], the "111" Project of China [B18022], the Fundamental Research Funds for the Central Universities [22221818014], the China Postdoctoral Science Foundation [2018M643243], the Natural Science Foundation from Shandong Province [ZR2017ZB0206], and the Shandong Taishan Scholar Award to Lixin Zhang.

## Author contributions

L.-X.Z. and G.-Y.T. conceived and supervised the project. L.-X.Z., G.-Y.T. and W.W. designed the experiments. M.L., Z.L and J.L. performed the experiments. M.L., W.W. and G.-Y.T. analyzed the data. G.Z, Z.W., and J.Y. performed the clinical sample test. L.L., L.K., M.W., K.W., Y.S., J.Y., L.Z., Z.Y., C.L., Q.Z., T.S., L.Z., F.X., H.D., X.L, J.Z., G.L, Y.Z., B.Z., C.L., S.L., X.X., Y.T. Y.L. and G.A. participated in this work. L.-X.Z. and G.-Y.T. wrote the manuscript.

## Additional information

**Competing interests:** The authors declare no competing interests.

