## [Peer Review File · Nature Communications]

Reviewers' comments:

Reviewer #1 (Remarks to the Author):

In this study, Liang et al developed a tool for rapid in vitro detection of small molecules using allosteric transcription factors and Cas12a proteins. ATF is attached to a carbohydrate binding module (CBM) and immobilized on microcrystalline cellulose. DsDNA contains both aTF binding site and a sequestered PAM. In the presence of the small molecule, aTF is released exposing the PAM. This mixture is incubated with Cas12a with a ssDNA quenched fluorophore. Binding to PAM, activates Cas12a which then cleaves ssDNA to generate a fluorophore signal. This is a neat idea for rapid cell-free detection of small molecules.

Here are my comments/concerns:

1. ATFs are generally dimers. Was each monomer attached independently to CBM? How did they ensure that both monomers were spatially close when attached to microcrystalline cellulose to be a functional dimer. This is different from EMSA assay of Fig. 3 which only tests free floating CBM-aTF fused protein.
2. The stoichiometry of aTF to dsDNA is a critical parameter because excess unbound dsDNA would give spurious signal in the next step (incubation with Cas12a). There was no discussion of optimization of this condition. They made it sound like it worked right off the bat which seems difficult to believe.
3. The above point is particularly important because different aTFs have different affinities for their DNA sequence. This means the condition has to be specifically optimized.
4. Other optimization parameters not discussed or not done are: increase in fluorescence as a function of dsDNA concentration. How sensitive is signal saturation to dsDNA concentration?
5. Two critical control time course experiments are missing (Fig. 4): (1) No ligand added to aTF tube. This will give background signal if no ligand were added (2) Unquenching of fluorophore over time with Cas12a present, no dsDNA added. This will give background activity of Cas12a.
6. The data has to be reported as Fold Change in signal (Fig. 4). In a real-life setting, only end point measurements are likely to be made. Unless, we know what fold change in signal indicates success, it is hard to assess the result. Fold change has to be relative to ratio of fluorescence with respect to both controls mentioned in the above (no ligand and background Cas12a).
7. It is not clear to me how they measured uric acid concentration using their tool in Figure 5 (X-axis). Did they use a calibration curve? Their tool only reports fluorescence changes. How was this converted to uric acid concentration. If they used a calibration curve, was it in the same medium (blood serum)?

Reviewer #2 (Remarks to the Author):

This article discusses the finding of the new detection method of uric acid using CaT-SMelor. The LOD of uric acid using CaT-SMelor is 10 nM and for pHBA is 1.8 nM. Correlation between CaT-SMelor and HPLC and biochemical analyser result are linear, this is a good correlation. Based on the result of the author done, I suggest a few data be added to make sure that this new detection method superior compare to HPLC and automatic biochemical analyzer. This article can be accepted after a few corrections.

My suggestions are :

1. The LOD of the HPLC method and Automatic biochemical analyzer need to be added into the article.
2. To make sure this new method superior, repeatability and reproducibility of the method need to be done as well as the HPLC method.
3. standard deviations of all measurement have to be added into the figure

Dr. Aliya Nur Hasanah

Reviewer #3 (Remarks to the Author):

In this paper, through combining the competitive binding activity of aTF for target DNA and Cas12a-based nucleic acid detection system, Liang et al. developed a new method (namely CaT-SMelor) for highly sensitive and specific detection of small molecules. Specifically, an aTF binds to target dsDNA in the absence of allosteric molecules, and thus prevents Cas12a from binding to the same target dsDNA. However, at the presence of certain allosteric small molecules, the aTF binds to the small molecules and releases from the target DNA, which then allows Cas12a to bind to the same target DNA and further triggers the Cas12a ssDNA trans-cleavage activity to produce fluorescence signals. Moreover, CaT-Smelor can detect target small compounds in a quantitative manner. In brief, this is a useful method for small molecule detection, which has the advantages of high speed, low-cost, simplicity and sensitivity. Therefore, I think the work has the merit to be published, and should be invited for revision.

Major concerns:

1. As the strong binding affinities of immobilized aTFs to target dsDNA is a critical factor of this detection system, I strongly recommend the authors to measure and compare the target dsDNA binding constants (Kd) of Cas12a-crRNA complex with that of immobilized aTFs (with and without the allosteric molecules).
2. Data related to Figure 5C are extremely important for judging the potential application scenarios of the method. Therefore, it is highly recommended that the data, which may include the number of samples tested, should be shown.
3. Error bars were not shown in the fluorescence intensity charts in Fig. 4, and should be added. Besides, the method for calculating the standard curve should also be provided.

Minor concerns:

1. The discussion part is a little bit lengthy and can be condensed.
2. In some cases, target compounds may be from complex sources, such as polluted water, food, fecal waste, etc., and the detection system can be damaged by the complexes, e.g. through disturbing or inactivating the enzymes. I think simple treatment of the samples may be needed in such situations, and the authors should discuss about this.
3. The word "CRISPR" were misspelled as "CIRSPR" in some places, including lines 30, 45, 291, 301 and 303.
4. It is highly recommend to use the singular form of "min" instead of the plural form of "mins" throughout the manuscript.
5. The drawback of the method and the point for further improvement may also be discussed. For example, the concentration of target molecules may need to be within a relatively narrow range when quantitated with this method.
6. In Table 1, it could be inappropriate to compare the nucleic acid detection methods with those of small molecules. In addition, more conventional methods (e.g. HPLC) for small molecule detection should be compared.

A point-by-point response to the reviewer(s)' comments

We would like to thank the reviewers for their helpful comments. Please find our responses below using blue color font.

Reviewer #1 (Remarks to the Author):

In this study, Liang et al developed a tool for rapid in vitro detection of small molecules using allosteric transcription factors and Cas12a proteins. ATF is attached to a carbohydrate binding module (CBM) and immobilized on microcrystalline cellulose. DsDNA contains both aTF binding site and a sequestered PAM. In the presence of the small molecule, aTF is released exposing the PAM. This mixture is incubated with Cas12a with a ssDNA quenched fluorophore. Binding to PAM, activates Cas12a which then cleaves ssDNA to generate a fluorophore signal. This is a neat idea for rapid cell-free detection of small molecules.

Here are my comments/concerns:

Reply: Thank you for your favorable comments. Specific answers to each question are listed as the follows:

Q1. ATFs are generally dimers. Was each monomer attached independently to CBM? How did they ensure that both monomers were spatially close when attached to microcrystalline cellulose to be a functional dimer. This is different from EMSA assay of Fig. 3 which only tests free floating CBM-aTF fused protein.

Reply: Yes, as you mentioned, aTFs are generally dimers; here we attached each monomer to CBD independently in our work. Both HosA and HucR belong to the multiple antibiotic resistance regulator (MarR) family of transcriptional regulators. Previous structure-based alignment reveals some conserved residues paly a critical role in dimerization for members of the MarR family. The two monomers interact with each other largely by hydrophobic interactions at the dimer interface through these conserved

hydrophobic residues (Mordelon et al, 2006, J Mol Biol. 2006, 360: 168-177). Therefore, this dimer is theoretically very stable in current aquatic buffer system, and not decomposing easily. Based on these considerations, as shown in following figure, either of cellules binding domain (CBD) attached to microcrystalline cellulose is probably suitable for our detection system.

Q2. The stoichiometry of aTF to dsDNA is a critical parameter because excess unbound dsDNA would give spurious signal in the next step (incubation with Cas12a). There was no discussion of optimization of this condition. They made it sound like it worked right off the bat which seems difficult to believe.

Reply: This is a very good comment. However, those unbound free dsDNA will be removed before we conduct the detection of target small molecules. We normally incubated the immobilized aTF proteins with saturated dsDNA so that all the DNA binding sites are occupied by dsDNA. Subsequently, the excess unbound free dsDNA was removed and washed twice (at least) with fresh buffer. To confirm the unbound free dsDNA was successfully removed via washing procedure, we co-incubated the last discarded supernatant with Cas12a containing fluorescence labeled ssDNA and observed no detected signal, which demonstrated that our procedure is sufficient to remove the unbound free dsDNA from the testing system.

Q3. The above point is particularly important because different aTFs have different affinities for their DNA sequence. This means the condition has to be specifically optimized.

Reply: We totally agree with the reviewer that it is very important to completely remove the unbound free dsDNA before do the detection.

It is quite right that different aTFs have different affinities for their DNA sequence. However, we are confident that the amount of dsDNA we proposed in our study is sufficient to saturate the DNA binding sites of aTFs we used. Because in our previous study, the binding affinities of immobilized aTFs to target dsDNA has been systematically characterized. We found that the number and position of point mutation in dsDNA could influence the equilibrium dissociation constant (K_D). Take HucR as an example, the corresponding sequence of dsDNA is TAGGTAGACATCTAAGTA (5' - 3'), and the value of K_D was 1.74 nM (Li et al, 2017, *Chem Common* 53, 99).

Q4. Other optimization parameters not discussed or not done are: increase in fluorescence as a function of dsDNA concentration. How sensitive is signal saturation to dsDNA concentration?

Reply: Thank the reviewer for this suggestion. As shown in Figure 4A, Figure S10 and Figure S11, the fluorescence intensity increases with the increase of dsDNA concentration. And signal saturation to dsDNA concentration is depended by aTF and dsDNA sequence. In current study, the selected aTFs (e.g. HucR, HosA) have been well-characterized in previous studies. In addition, for the application of an uncharacterized aTF proteins, some routine but important work, e.g. the binding affinity toward different dsDNA, selection of suitable dsDNA (sequence), allosteric activity, should be performed in advance.

Q5. Two critical control time course experiments are missing (Fig. 4): (1) No ligand added to aTF tube. This will give background signal if no ligand were added (2) Unquenching of fluorophore over time with Cas12a present, no dsDNA added. This will give background activity of Cas12a.

Reply: Thank the reviewer for pointing this out. Actually, in Figure 4, the background signal (no ligand added to aTF tube) has already been subtracted, we have added this information in figure legend in the revised version. In this work, the fluorophore-quencher (FQ) double-labeled short ssDNA probe has been applied. Normally, there was no fluorescent signal because the fluorescent group (5-FAM) is quenched by the quencher group (BHQ) (Chen et al, 2018, Science, 360, 436). Only in the presence of target free dsDNA and Cas12a (including crRNA), FQ-labeled ssDNA probe could be cleavage by activated Cas12a, and then produce fluorescent signal.

Q6. The data has to reported as Fold Change in signal (Fig. 4). In a real-life setting, only end point measurements are likely to be made. Unless, we know what fold change in signal indicates success, it is hard to assess the result. Fold change has to be relative to ratio of fluorescence with respect to both controls mentioned in the above (no ligand and background Cas12a).

Reply: Thank the reviewer for this suggestion. According to current CaT-SMelor method, target small molecules can be qualitatively or quantitatively analyzed by measuring the change in the fluorescent signal. During the qualitative detection of target small molecules, the fluorescence signal was measured without (control) or with small molecules, as reviewer's suggestion. The results indeed could be presented as fold change in signal. For quantitative detection of target small molecules, the time-course of fluorescence intensity changes should be measured. And the slope of fluorescence linear region between 5-20min (normally) actually represent a ssDNA trans-cleavage rate of CRISPR/Cas12a. Our study further indicated that the trans-cleavage rate (or slope) and small molecule concentration (in an appropriate range) show a very good linear relationship. This is the reason why current CaT-SMelor method also can be applied for quantitative detection of target small molecules. In addition, we found that

the background signal (control, no ligand) was very stable and can be considered as a constant. Therefore, both two data processing modes (deduct background or present as fold change) are suitable for current CaT-SMelor method.

Q7. It is not clear to me how they measured uric acid concentration using their tool in Figure 5 (X-axis). Did they use a calibration curve? Their tool only reports fluorescence changes. How was this converted to uric acid concentration. If they used a calibration curve, was it in the same medium (blood serum)?

Reply: Yes, in present study, the uric acid concentration in samples were quantitative analysis by calibration curve. As mentioned above (Reply for Q6), for calibration curve construction, after incubation with different concentrations of uric acid standard, the free dsDNA dissociated from aTF (immobilized)-dsDNA complex were detected with the Cas12a-based FQ-labeled reporter system. The time-course of fluorescence intensity changes and the slope of fluorescence linear region between 5-20min (normally) show a uric acid concentration dependent manner (Figure 5, Figure S12), i.e., the trans-cleavage rate (or slope) and uric acid standard concentration (in an appropriate range) show a linear relationship. For serum uric acid analysis, each serum sample was divided into 3 additional tubes, and then quantified by HPLC, CaT-SMelor and clinical automatic biochemical analyzer system (Beckman kit), respectively. Please also see line 448-456 and Figure S15.

Reviewer #2 (Remarks to the Author):

This article discusses the finding of the new detection method of uric acid using CaT-SMelor. The LOD of uric acid using CaT-SMelor is 10 nM and for pHBA is 1.8 nM. Correlation between CaT-SMelor and HPLC and biochemical analyser result are linear, this is a good correlation. Based on the result of the author done, I suggest a few data be added to make sure that this new detection method superior compare to

HPLC and automatic biochemical analyzer. This article can be accepted after a few corrections.

My suggestions are:

Reply: We appreciate the comments and suggestions from the reviewer. Specific answers to each question are listed as follow:

Q1. The LOD of the HPLC method and Automatic biochemical analyzer need to be added into the article.

Reply: Thank the reviewer for this suggestion. In our study, the limit of detection (LOD) of the HPLC for uric acid was 0.5~1 μM . According to manufacture instruction, the LOD of automatic biochemical analyzer (Kit number 442785, Beckman Coulter, Inc.) for uric acid was 30 μM . As reviewer's suggestion, this date has been added into Table 1.

Q2. To make sure this new method superior, repeatability and reproducibility of the method need to be done as well as the HPLC method.

Reply: Thanks for reviewer's suggestion. To evaluate the repeatability and reproducibility of the CaT-SMelor, four serum samples have been randomly selected for uric acid analysis. As shown in following figure and table, different serum samples (S1, S2, S3, S4) are independently detected by CaT-SMelor for one time (with three repeats) or twice (with two repeats each). the linear trend lines of fluorescence signal induced by different serum samples, and the slope of fluorescence intensity has been calculated. And it was found that the data obtained in current study shows good consistency and reproducibility. Please also see Figure 5 and source data.

Serum sample	Experiment 1				Experiment 2		
	Repeat 1	Repeat2	Repeat3	SD%	Repeat 1	Repeat2	SD%
S1	2883.2	2939.4	N/A	1.3%	2861.1	2686.2	4.4%
S2	3221.3	2949	N/A	4%	2921.9	2803.2	2.9%
S3	2302.1	2243.7	2325.6	1.8%	N/A	N/A	N/A
S4	1854	1756.8	1794.1	2.7%	N/A	N/A	N/A

Q3. Standard deviations of all measurement have to be added into the figure.

Reply: Thank the reviewer for this suggestion. The standard deviations were added into the revision.

Please see Figure 5 and source data.

Reviewer #3 (Remarks to the Author):

In this paper, through combining the competitive binding activity of aTF for target DNA and Cas12a-based nucleic acid detection system, Liang et al. developed a new method (namely CaT-SMelor) for highly sensitive and specific detection of small molecules. Specifically, an aTF binds to target dsDNA in the absence of allosteric molecules, and thus prevents Cas12a from binding to the same target dsDNA. However, at the presence of certain allosteric small molecules, the aTF binds to the small molecules and releases from the target DNA, which then allows Cas12a to bind to the same target DNA and further triggers the Cas12a ssDNA trans-cleavage activity to produce fluorescence signals. Moreover, CaT-Smelor can detect target small compounds in a quantitative manner. In brief, this is a useful method for small molecule detection, which has the advantages of high speed, low-cost, simplicity and sensitivity. Therefore, I think the work has the merit to be published, and should be invited for revision.

Reply : Thank you for your favorable comments. We have fully addressed all the comments, and listed the point-by-point response as follow.

Major concerns:

Q1. As the strong binding affinities of immobilized aTFs to target dsDNA is a critical factor of this detection system, I strongly recommend the authors to measure and compare the target dsDNA binding constants (K_d) of Cas12a-crRNA complex with that of immobilized aTFs (with and without the allosteric molecules).

Reply: Thank the reviewer for this suggestion. We agree that the strong binding affinities of the aTFs to target dsDNA is a critical factor of this detection system. In our previous study, the binding affinities of immobilized aTFs to target dsDNA has been systematically characterized. We found that the number and position of point mutation in dsDNA could influence the equilibrium dissociation constant (K_D). Take HucR as an example, the corresponding sequence of dsDNA is TAGGTAGACATCTAAGTA (5' - 3'), and the

value of K_D was 1.74 nM (Li et al, 2017, *Chem Commun* 53, 99).

The specific binding of uric acid (UA)–HucR and 4-HBA–HosA also have been reported (Roy et al, 2016, *Biochemistry* 55, 1120), but their K_D values were not given. To provide the accurate thermodynamic data, we characterized the K_D values of UA–HucR and 4-HBA–HosA using isothermal titration calorimetry (Cao et al, 2018, *Sci Adv* 4, eaau4602). They are $9.35 \pm 0.99 \mu\text{M}$ and $4.74 \pm 0.25 \mu\text{M}$, respectively. When the small molecule signal presents, the affinity of aTF for DNA usually reduce several orders of magnitude. For example, tetracycline induces conformational changes of TetR that reduce the binding constant of TetR and DNA by 6–9 orders of magnitude (Orth et al, 2000, *Nat Struct Biol* 7, 215; Saenger et al, 2000, *Angew Chem Int Ed* 39, 2042).

In this study, when we compared the dsDNA-binding and allosteric activities of the original HosA (Fig.S5), there were no significant differences between HosA and CBD-HosA on gel-shift images or in the percentage of dsDNA shifted (Fig. 2D–2E & Fig. S3). These results indicated that the fusion of CBD (immobilization) had no significant effect on the functions of the selected aTFs.

In addition, the target dsDNA binding constants of Cas12a-crRNA complex couldn't be directly detected, because in our study, the dsDNA will be cleavage by Cas12a-crRNA complex. However, by using inactive dCas9 (D10A; H840A), previous study indicated that the binding affinity of dCas9-gRNA for dsDNA was 0.036 nM (O'Connell et al, 2014, *Nature* 516, 263).

Q2. Data related to Figure 5C are extremely important for judging the potential application scenarios of

the method. Therefore, it is highly recommended that the data, which may include the number of samples tested, should be shown.

Reply: Thanks to your suggestion. Please see Table S1 in the revision.

Q3. Error bars were not shown in the fluorescence intensity charts in Fig. 4, and should be added. Besides, the method for calculating the standard curve should also be provided.

Reply: We accepted this suggestion. Error bars could be found in source data. The method for calculation of the standard curve has been provided. Please see Figure S15 in the revision.

Minor concerns:

Q4. The discussion part is a little bit lengthy and can be condensed.

Reply: The discussion section has been updated as you suggested. Please see line 290-298, 314-316, 318-319 in the revision.

Q5. In some cases, target compounds may be from complex sources, such as polluted water, food, fecal waste, etc., and the detection system can be damaged by the complexes, e.g. through disturbing or inactivating the enzymes. I think simple treatment of the samples may be needed in such situations, and the authors should discuss about this.

Reply: Thank the reviewer for this suggestion. We agree that the minimal sample preparation, such as isolation or extraction, is necessary for any of detection methods. In addition, the detection system damaged by sample (containing or contaminated by, eg. denaturant, organic solvent, heavy metals), which are general issues for most of protein or enzyme -based bio-detection platform, include but no

limited to current method.

Q6. The word “CRISPR” were misspelled as “CIRSPR” in some places, including lines 30, 45, 291, 301 and 303.

Reply: Thank the reviewer for pointing this out. We corrected accordingly in the revision. Sorry for our mistake. These misspelled words have been corrected.

Q7. It is highly recommended to use the singular form of “min” instead of the plural form of “mins” throughout the manuscript.

Reply: Thank the reviewer for pointing this out, we corrected in the revision.

Q8. The drawback of the method and the point for further improvement may also be discussed. For example, the concentration of target molecules may need to be within a relatively narrow range when quantitated with this method.

Reply: Thank the reviewer for this suggestion. The limitations of the method have been added to the discussion, and our previous study indicated that the linear detection range of CaT-SMelor for targeted small molecules can be tuned by changing the sequence of the dsDNA (Li et al, 2017, *Chem Common* **53**, **99**). Please see line 350-352.

Q9. In Table 1, it could be inappropriate to compare the nucleic acid detection methods with those of small molecules. In addition, more conventional methods (e.g. HPLC) for small molecule detection should be compared.

Reply: Thank the reviewer for this suggestion. The principle underlying this method is that the unreadable signals of the small molecule targets are transformed into easily readable dsDNA signals by aTFs. As reviewer's suggestion, HPLC as a very conventional method for small molecule detection has been added into Table 1.

REVIEWERS' COMMENTS:

Reviewer #1 (Remarks to the Author):

Concerns were addressed.

I would suggest that the authors briefly discuss key parameters that have to be tuned to implement their technique for other allosteric TFs.

Reviewer #3 (Remarks to the Author):

The revised manuscript has been much improved, and all my previous concerns have been fully addressed. Therefore, I believe the present manuscript is suitable for publishing in NC.

REVIEWERS' COMMENTS:

Reviewer #1 (Remarks to the Author):

Concerns were addressed.

I would suggest that the authors briefly discuss key parameters that have to be tuned to implement their technique for other allosteric TFs.

Reply: According to this study, we found that the CRISPR/Cas12 -mediated downstream nucleic acid detection platform was really a supersensitive and highly faithful general signal output and display module. Suppose we plan to implement this technique for other allosteric TFs, the aTF -mediated upstream signal transfer/input module should be well characterized and optimized. And the most important parameters should be tuned are the dsDNA binding affinity and allosteric activity of selected aTFs. Our previous studies indicated that reducing the affinity (in an appropriate range) between the aTF and dsDNA (increasing K_D) or increasing the affinity between the aTF and targeted small molecular (reducing K_I) can improve the sensitivity and broaden the linear ranges of biosensor system. The K_D value could be easily tuned by changed the sequence of dsDNA or introduce selective nick sites within dsDNA (Li et al, Chem Commun, 2017; 53:99; Cao et al, Sci Adv, 2018; 4: eaau4602). An ideal mutant of aTF with both increased K_D and reduced K_I may require some protein engineering work, such as phage-assisted continuous evolution.